# Research on Color Image Encryption Algorithm Based on Bit-Plane and Chen Chaotic System

**DOI:** 10.3390/e24020186

**Published:** 2022-01-26

**Authors:** Jiangjian Xu, Bing Zhao, Zeming Wu

**Affiliations:** Electronic Engineering College, Heilongjiang University, Harbin 150080, China; 2211746@s.hlju.edu.cn (J.X.); 2201669@s.hlju.edu.cn (Z.W.)

**Keywords:** bit plane, Chen chaotic system, logistic chaotic system, image encryption

## Abstract

In response to the problems of high complexity and the large amount of operations of existing color image encryption algorithms, a low-complexity, low-operation color image encryption algorithm based on a combination of bit-plane and chaotic systems is proposed that is interrelated with plaintext information. Firstly, three channels of an RGB image are extracted, and the gray value of each pixel channel can be expressed by an eight-bit binary number. The higher- and lower-four bits of the binary gray value of each pixel are exchanged, and the position of each four-bit binary number is scrambled by a logistic chaotic sequence, and all the four-bit binary numbers are converted into hexadecimal numbers to reduce the computational complexity. Next, the position of the transformed image is scrambled by a logistic chaotic sequence. Then, the Chen chaos sequence is used to permute the gray pixel values of the permuted image. Finally, the gray value of the encrypted image is converted into a decimal number to form a single-channel encrypted image, and the three-channel encrypted image is synthesized into an encrypted color image. Through MATLAB simulation experiments, a security analysis of encryption effects in terms of a histogram, correlation, a differential attack, and information entropy is performed. The results show that the algorithm has a better encryption effect and is resistant to differential attacks.

## 1. Introduction

With the rapid development of network information technology, the security of information transmission has become a growing concern. Digital images, as a common carrier for information transmission, are widely used in various fields such as medical, military, industrial, and daily life. However, digital images are characterized by a large amount of information, high redundancy, and strong correlation, making common encryption algorithms unable to meet the requirements of image encryption. For example, there is the Data Encryption Standard (DES), the International Data Encryption Algorithm (IDEA), and the Advanced Encryption Standard (AES) [1]. Therefore, many algorithms concerning image encryption have been proposed; for example, optical information processing technology has advantages in the field of image encryption, with its fast speed and high degree of freedom [2]; the random grid (RG) algorithm forms ciphertext by encrypting an image into multiple shared images and then superimposing the shared images [3,4]; DNA coding [5,6] has ultra-low power consumption, large storage capacity, and parallel computing features have advantages in image encryption; the compression-aware (CS) algorithm encrypts images by sampling and reconstructing them [7]; chaos theory, with characteristics such as ergodicity, randomness, and initial value sensitivity due to chaotic systems with deterministic but unpredictable states of motion, has made the study of chaotic systems for digital image encryption algorithms increasingly popular [8,9,10]. At the same time, in the process of network information transmission, color images carry richer information. Therefore, the study of color image encryption algorithms has become a hot research topic [11,12]. Kaur et al. [13] proposed a five-dimensional hyperchaotic system and a non-dominated sequential genetic approach to local chaos searches for encrypting color images. Tong et al. [14] proposed a four-dimensional chaotic-based system and designed a new pseudo-random sequence generator for the fast encryption of color images. Ran et al. [12] proposed a hyperchaotic Lorenz-based chaotic system and encrypted quantum color images. Zhou et al. [15] proposed a new one-dimensional chaotic system by combining existing one-dimensional chaotic unions and encrypting color images.

The encryption of digital images by chaotic sequences is generally accomplished by means of scrambling and diffusion [16]. Scrambling is achieved by changing the position of the pixels to achieve an encrypted effect. Diffusion achieves the function of encryption by changing the grayscale value of the pixels. Liu et al. [17] proposed an encryption algorithm based on a bit-plane and a modified logistic chaos system. The digital image is, first, globally scrambled, and then the scrambled image is divided into four high-bit planes and four low-bit planes, and the four low-bit planes are used to generate a chaotic sequence with the chaotic system and perform an aliasing calculation on the high-bit planes to finally form an encrypted image. However, this method does not perform pixel diffusion on the four lower planes of the image, and there is a certain security risk. Li et al. [18] used a slant tent system to generate two alternative control sequences for diffusion between bit planes. A Rucklidge system was used to generate a position-shifted control sequence of pixels and a bit-period-shifted control sequence for bit-level permutation. However, the selection of the initial value of the chaotic system relies on the input of an external key, which cannot vary with the image. Wang et al. [19] proposed a new heterogeneous bit-alignment correlated chaotic color image encryption algorithm. The heterogeneous position is used to reduce the computational effort by exploiting the difference in the amount of information between bit planes. However, each scrambling operation can only disrupt the bit positions between two bit planes at the same time, and cannot scramble the positions of the bits between multiple bit planes at the same time. Panwar et al. [20] used a combination of DNA coding and multiple one-dimensional chaotic sequences to encrypt the images. However, the computational complexity of the encryption algorithm is high.

Therefore, in order to overcome the shortcomings in the above literature, this paper proposes a color image encryption algorithm based on bit-plane information, which is associated with plaintext images with high security and low complexity. The algorithm turns a pixel value into two hexadecimal numbers when encrypting the image, reducing the computational complexity of the encryption process. Moreover, the parameters and initial values of the chaotic system used in the encryption process for each channel in the color image are related not only to the plaintext information of the corresponding channel, but also to the plaintext information of the other two channels, making the connection between the plaintext and the key stronger and the key space larger. Finally, using MATLAB simulation, the experimental results show that the algorithm has a good encryption effect and can resist various attacks.

The rest of the paper is organized as follows. Firstly, the basic theories used in encryption algorithms are introduced, including bit-transformation theory and chaos theory. Next, the process of encryption and decryption algorithms is described in detail. Next, the experimental simulation results and safety analysis are presented. Then, the discussion is introduced. Finally, the conclusion to the paper is presented.

## 2. Basic Theory

Image encryption is widely used in industry, medicine, and daily life [8,9], but the requirements for image encryption in various fields are getting higher and higher, with very high security and very low computational complexity. Chen chaotic systems are less complex than high-dimensional chaotic systems, but much more complex than low-dimensional chaotic systems. Therefore, it can be chosen as an encryption sequence.

### 2.1. Bit Plane

The pixel values of a grayscale image range from 0 to 255, and by converting the grayscale values to binary, we can obtain an eight-bit binary number, which ranges from 000000 to 11111111. Then, each image with the same position of binary grayscale can form eight images of the same size as the original image, and such images are called bit planes. A grayscale image can be divided into eight bit planes, but each bit plane has a different weight and, therefore, the amount of information in each bit plane varies. The proportion of information in each bit-plane can be expressed by Equation (1), and the bit-plane information map of the Lena image is shown in Figure 1.
(1)$$I\left(i\right)=\frac{2^{i-1}}{255}\;\;i=1,\;2,\;3,\;4,\;5,\;6,\;7,\;8$$
where *i* represents the *i*th plane and I(*i*) represents the proportion of information in the *i*th plane. The weights and percentages of each bit plane are shown in Table 1.

From Table 1, it can be seen that the higher the number of bits in the bit plane, the higher the proportion of information, and that the lower the number of bits in the bit plane, the lower the proportion of information. The sum of the information volume of the higher-four bit planes is 94.118%, and the sum of the information volume of the lower-four bit planes is 5.882%. Therefore, the amount of information in a grayscale image is mainly concentrated in the bit plane of the higher-four bits.

As can be seen from Figure 1, the higher the number of bits in the bit plane, the greater the proportion of information, especially the eighth bit plane, which accounts for the largest proportion of information. The lower-four bit planes are completely invisible in the original image, and the proportion of information is very small. Therefore, the difference in the proportion of information in the bit planes can be used to encrypt the image. There is a large difference in the proportion of information between the high and low bit planes, but encryption of the low bit planes is also necessary because it can be used to resist selective plaintext attacks.

### 2.2. Chaotic Sequences

#### 2.2.1. Logistic Chaotic Sequences

Logistic chaotic systems are the most widely used one-dimensional discrete chaotic systems, characterized by nonlinear dynamics, and their mapping equations are:(2)$$x_{n+1}=\mu x_{n}\left(1-x_{n}\right)$$
where μ ∈ (0,1), $x_{n}$∈ (0,1). When 3.5699 < μ ≤ 4, the system is in a state of chaos. When μ = 4, the complexity of the chaotic system is greatest. When μ = 3.99, $x_{n}$ = 0.555, and 0.556; the scatter plot of the logistic chaotic system is shown in Figure 2.

As can be seen from Figure 2, the chaotic sequences are scattered irregularly and have random-like properties. The two sequences overlap at the beginning, when there is a small change in the initial value, but later, the two sequences spread out and have initial value sensitivity. However, when using logistic chaotic sequences, it is necessary to iterate over the chaotic sequences and discard the previous sequence in order to eliminate transient effects.

#### 2.2.2. Chen Chaotic Sequences

The Chen chaotic system was proposed by Professor Guanrong Chen in his research on the inverse control of chaos, which has similarities with the Lorenz chaos system but is a little more complex in terms of phase space. Many scholars have used the Chen chaos system to encrypt images [21,22], and the mathematical model is:(3)$$\left\{\begin{array}{c} \dot{x}=a\left(y-x\right)\;\;\;\;\;\;\;\;\;\\ \dot{y}=\left(c-a\right)x-xz+cy\\ \dot{z}=xy-bz\;\;\;\;\;\;\;\;\;\; \end{array}\right.$$
where *x*, *y*, *z* is the state quantity of the system, and *a* > 0, *b* > 0, *c* > 0 are the parameters of the system. When *a* = 35, *b* = 3, *c* = 28, the system can exhibit a chaotic state; the simulation results are shown in Figure 3.

As can be seen from Figure 3, the Chen chaotic system is highly complex and unpredictable in space, but the chaotic behaviour is all in a finite space and each curve is not tangential. Thus, it is possible to encrypt images using Chen chaotic sequences, and using the Chen chaotic system has some advantages: the complexity of this chaotic system is higher than that of the low-dimensional chaotic system, and the three components of the three-dimensional chaotic system correspond exactly to the three channels R, G, and B in the color image.

## 3. The Proposed Encryption and Decryption Algorithms

### 3.1. Encryption Algorithm

For the encryption of a color image of the M × N × 3, the encryption scheme flow is shown in Figure 4.

The color image is divided into three channels, R, G, and B, and three grayscale images are obtained. The internal keys K1, K2, and K3 and the key Kr are calculated based on the plaintext information. Then, the initial values and parameters of the logistic chaos system and the Chen chaos system are calculated from the keys and iterated to form a chaotic sequence. The binary images Pr1, Pg1, and Pb1 are formed by converting the grayscale values of the pixels of the three channels R, G, and B into eight-bit binary numbers of size 1 × 8MN. The binary images Pr3, Pg3, and Pb3 are obtained by permuting the positions of 0 and 1 in the high and low quarters of the binary numbers using a logistic chaotic sequence. The high and low quarters of the binary numbers are converted into hexadecimals to obtain hexadecimal images Pr4, Pg4, and Pb4 of size 1 × 2MN. The images Pr5, Pg5, and Pb5 are obtained by permuting the pixel positions of the hexadecimal image with a logistic chaos sequence. The images Pr6, Pg6, and Pb6 were obtained by diffusing the pixel values of Pr5, Pg5, and Pb5 using the Chen chaos sequence. Finally, the encrypted images of the three channels are combined to form the color encrypted image P1.

#### 3.1.1. Calculation of Internal Keys K1, K2, K3, and Key Values Kr

The gray value of the adjacent two rows of pixels of the R channel image is the XOR operation used to obtain $k_{i}$. Then, the average value is calculated and rounded to obtain $k_{i}{}^{\prime }$. Next, the data in the $k_{i}{}^{\prime }$ is divided into 20 groups; the data amount of each group is P. The average value of each group is calculated separately and integerized, and finally, the internal key *K*1 of the R-channel image is obtained. The internal key *K*1 is shown in Equation (4). Similarly, the internal keys *K*2 and *K*3 of the G and B channels can be derived. Use *K*1, *K*2, and *K*3 to XOR each other to get *Kr*′, and finally, calculate the average of *Kr*′ and round it up to get *Kr*. The key value *Kr* is shown in Equation (5):(4)$$\{\begin{array}{c} k_{i}=Pr\left(i,1:M\right)Pr\left(i+1,M\right)\;\;\;i=1,\;2,\;3\ldots ,\;M-1\;\;\;\;\;\;\;\;\;\;\;\;\;\;\;\;\\ k_{i}^{{}^{\prime }}=floor\left(\frac{1}{M}\overset{M}{\underset{j=1}{\sum }}k_{i}\left(j\right)\right)\;\;\;\;\;\;\;i=1,\;2,\;3,\ldots ,M-1\\ P=floor\left(\frac{M-1}{20}\right)\\ K1\left(q\right)=floor\left(\frac{1}{P}\overset{qP}{\underset{i=1+\left(q-1\right)P}{\sum }}k_{i}^{{}^{\prime }}\right)\;\;\;\;\;\;\;q=1,\;2,\;3,\ldots ,\;20\; \end{array}$$
(5)$$\left\{\begin{array}{c} Kr^{\prime }=K1⊕K2⊕K3\;\;\;\;\;\;\;\;\;\\ Kr=floor(\frac{1}{20}\overset{20}{\underset{i=1}{\sum }}Kr{}^{\prime }\left(i\right))\;\;\; \end{array}\right.$$
where ⊕ represents an XOR operation. $Pr\left(i,1:M\right)$ represents the pixel value of the *i*th row in the R-channel image. *floor* () represents a down integer operation. The resulting internal key *K*1 consists of 20 integers between 0 and 255.

#### 3.1.2. Formation of Logistic Chaotic Sequences and Chen Chaotic Sequences

Take an R-channel image as an example: the initial values *X*1(0) and *X*2(0) and the parameters μ1 and μ2 are calculated for the two logistic chaotic sequences, as shown in the Equation (6). The initial values of the Chen chaotic system are calculated *x*(0), *y*(0), and *z*(0), as shown in Equation(7):(6)$$\{\begin{array}{c} X1\left(0\right)=\frac{1}{256}(floor(\frac{1}{MN}\sum_{i=1}^{M}\sum_{j=1}^{N}Pr\left(i,j\right))⊕Kr)+\frac{1}{256}(\frac{1}{MN}\sum_{i=1}^{M}\sum_{j=1}^{N}Pr\left(i,j\right))*10^{-5}\\ \mu 1=3.9+\frac{1}{256}\left(K1\left(1\right)⊕K1\left(2\right)⊕K1\left(3\right)⊕K1\left(4\right)⊕K1\left(5\right)⊕Kr\right)*0.1\\ X2\left(0\right)=X1\left(0\right)+\frac{1}{256}(K1\left(11\right)⊕K1\left(12\right)⊕K1\left(13\right)⊕K1\left(14\right)⊕K1(15)⊕Kr)*10^{-10}\\ \mu 2=3.9+\frac{1}{256}\left(K1\left(6\right)⊕K1\left(7\right)⊕K1\left(8\right)⊕K1\left(9\right)⊕K1\left(10\right)⊕Kr\right)*0.1\;\;\; \end{array}$$
(7)$$\{\begin{array}{c} aver=\frac{1}{MN}\sum_{i=1}^{M}\sum_{j=1}^{N}Pr\left(i,j\right)\;\;\;\;\;\;\;\;\;\;\;\;\;\;\;\;\;\;\;\;\;\;\;\;\;\;\;\;\;\;\;\;\;\;\;\;\;\;\;\;\;\;\;\;\;\;\;\;\;\\ x\left(0\right)=\frac{1}{256}\left(K1\left(1\right)⊕K1\left(4\right)⊕K1\left(7\right)⊕K1\left(10\right)⊕Kr⊕floor\left(aver\right)\right)+1+\frac{aver}{256}*10^{-5}\\ y\left(0\right)=\frac{1}{256}\left(K1\left(2\right)⊕K1\left(5\right)⊕K1\left(8\right)⊕K1\left(11\right)⊕Kr⊕floor\left(aver\right)\right)+2+\frac{aver}{256}*10^{-5}\\ z\left(0\right)=\frac{1}{256}\left(K1\left(3\right)⊕K1\left(6\right)⊕K1\left(9\right)⊕K1\left(12\right)⊕Kr⊕floor\left(aver\right)\right)+3+\frac{aver}{256}*10^{-5} \end{array}$$
where aver represents the average of the grayscale values of the R-channel image.

The logistic and Chen chaotic sequences were iterated (*M* + *K*(16) + *K*(17) + *K*(18) + *K*(19) + *K*(20) + *Kr*) times, and were used to eliminate the adverse effects of transient effects. At the end of the iteration, the logistic chaos sequence *X*1 of length 8MN is obtained by iterating 8MN times again, and the logistic chaos sequence *X*2 of length 2MN and the Chen chaos sequences *x*, *y*, and *z* of length 2MN, respectively, are obtained after iterating 2MN times.

#### 3.1.3. Bit Transformation

Take the R channel as an example: the pixel value of the grayscale image Pr1 ranges from 0 to 255, and the gray value is converted to a binary number, that is, the pixel value ranges from 000000000 to 11111111, and a binary image with a size of 1 × 8MN is obtained. The binary image Pr2 of size 1 × 8MN is obtained by swapping the higher- and lower-four bits of the binary number in Pr1, as shown in Equation (8):(8)$$\;\mathrm{Pr}2t=\left\{\begin{array}{c} \mathrm{Pr}1\left(mod\left(t,8\right)+4+8floor\left(\frac{t-1}{8}\right)\right)\;mod\left(t,8\right)=1,\;2,\;3,\;4\\ \mathrm{Pr}1\left|mod\left(t,8\right)-4+8floor\left(\frac{t-1}{8}\right)\right|\;\;mod\left(t,8\right)=5,\;6,\;7,\;0 \end{array}\right.t=1,\;2,\;3,\ldots ,\;8\mathrm{MN}$$
where Pr2t represents the *t*-th element in the pr2 image. *mod*() indicates the remainder operation. The image with a size of 1 × 8MN is divided into groups for each of the eight elements, and the positions of the first four elements and the last four elements in each group are interchanged, resulting in an image with a size of 1 × 8MN Pr2.

Each of the four elements in the logistic chaotic sequences $x1_{i}$ and $\mathrm{Pr}2_{i}$ is divided into a group, and each group is denoted as $x1_{i}$, $\mathrm{Pr}2_{i}$. Then, the elements in the $x1_{i}$ are ascended to obtain the index sequence $s_{i}$, and then the $\mathrm{Pr}2_{i}$ is scrambled with the index sequence $s_{i}$ to obtain the $\mathrm{Pr}3_{i}$. Finally, all the $\mathrm{Pr}3_{i}$ are combined to form Pr3, as shown in Equation (9):(9)$$\left\{\begin{array}{c} \left[p\;s_{i}\right]=sort\left(x1_{i}\right)\;\;\;\;\;\;\;\;\;\;\;\;\;\\ \mathrm{Pr}3_{i}=\mathrm{Pr}2_{i}\left(s_{i}\right)\;\;\;\;\;\;\;\;\;\;\;\;\;\;\;\\ \mathrm{Pr}3=\left[\mathrm{Pr}3_{1},\mathrm{Pr}3_{2},\ldots ,\mathrm{Pr}3_{2\mathrm{MN}}\right]\;\; \end{array}\;\;\;\right.i=1,\;2,\ldots ,\;2\mathrm{MN}$$
where *sort*() represents an ascending operation, *p* is the ascending sequence, and $s_{i}$ is the index value of the ascending sequence.

Each of the four elements of Pr3 is divided into groups of four binary digits, each of which has a value in the range 0000 to 1111. All four binary digits are converted to hexadecimals, and each digit has a value in the range 0 to 15, resulting in an image Pr4 of size 1 × 2MN.

#### 3.1.4. Pixel Position Is Scrambled

The logistic chaotic sequence *X*2 is subjected to an ascending operation to obtain the index sequence index, and then the index sequence index is used to permute Pr4 to obtain the image Pr5 of size 1 × 2MN, as shown in Equation (10):(10)$$\left\{\begin{array}{c} \left[B\;index\right]=sort\left(X2\right)\\ \mathrm{Pr}5=\mathrm{Pr}4\left(index\right)\;\;\; \end{array}\right.\;\;\;\;i=1,\;2,\;3,\ldots ,\;2\mathrm{MN}$$

#### 3.1.5. Pixel Gray Value Diffusion

Pixel gray value diffusion is to change the size of the pixel gray value to achieve the purpose of encryption. This article uses three operation methods, an XOR operation, an addition operation, and a subtraction operation, taking the logistic chaotic sequence *X*1 with a length of 8MN, intercepting the first 2MN elements as the new sequence *x*3, and then quantizing it to obtain the sequence *X*3, as shown in Equation (11). The Chen chaotic sequence is also quantified to obtain new sequences *X*, *Y*, and *Z*, as shown in Equation (12):(11)$$X3=mod\left(floor\left(x3*10^{16}\right),3\right)$$
(12)$$\left\{\begin{array}{c} X=mod\left(floor\left(x*10^{10}\right),16\right)\\ Y=mod\left(floor\left(y*10^{10}\right),16\right)\\ Z=mod\left(floor\left(z*10^{10}\right),16\right) \end{array}\right.$$
where all elements in *X*3 are permutations of 0, 1, and 2. The elements in *X*, *Y*, and *Z* are all permutations of integers between 0 and 15. Among them, *X* is used to diffuse pixel gray values for the R-channel image, *Y* is used to diffuse pixel gray values for the G-channel image, and *Z* is used to diffuse pixel gray values for the B-channel image. Take the R channel as an example: if *X*3(*i*) = 0, then perform the XOR operation on Pr5(*i*) and *X*(*i*); if *X*3(*i*) = 1, then perform the addition operation on Pr5(*i*) and *X*(*i*); if *X*3(*i*) = 2, then subtract Pr5(*i*) and *X*(*i*). Finally, the encrypted image Pr6, after grayscale diffusion, is obtained. The calculation is shown in Equation (13):(13)$$\left\{\begin{array}{c} \mathrm{Pr}6\left(i\right)=f_{⊕}\left(\mathrm{Pr}5\left(i\right),X\left(i\right)\right)\;\;\;\;X3=0\\ \mathrm{Pr}6\left(i\right)=f_{+}\left(\mathrm{Pr}5\left(i\right),X\left(i\right)\right)\;\;\;\;X3=1\\ \mathrm{Pr}6\left(i\right)=f_{-}\left(\mathrm{Pr}5\left(i\right),X\left(i\right)\right)\;\;\;\;X3=2 \end{array}\right.\;\;i=1,\;2,\ldots ,\;2\mathrm{MN}$$

The hexadecimal of each two bits in Pr6 is converted to a decimal to obtain an image Pr7 of size 1 × MN, and then Pr7 is reshaped into an M × N image. Finally, the encrypted images of the three channels, R, G, and B, are combined into a color encrypted image P1.

### 3.2. Decryption Algorithm

The decryption process and the encryption process are mutually inverse processes. The keys, chaotic sequences, and bit transformations used in the decryption process are the same as in Section 3.1.1, Section 3.1.2 and Section 3.1.3. Additive and subtractive operations are used in the pixel gray value substitution during encryption, and during decryption, the additive operations are changed to subtractive operations and the subtractive operations are changed to additive operations. The rest of the decryption process is the inverse of the encryption process. A block diagram of the decryption scheme is shown in Figure 5.

The ciphertext image is divided into three channels, R, G, and B, and three grayscale images are obtained. The initial values and parameters of the logistic chaos system and the Chen chaos system are calculated by the key and iterated to form the chaotic sequence. The grayscale values of the R, G, and B channel image pixels are converted into two-bit hexadecimal numbers to form binary images Pr6, Pg6, and Pb6 of size 1 × 2MN. The pixel values of Pr6, Pg6, and Pb6 are inverted with the Chen chaos sequence to obtain images Pr5, Pg5, and Pb5. The hexadecimal images Pr4, Pg4, and Pb4 are obtained by inverse permutation of the pixel positions of the images Pr5, Pg5, and Pb5 using a logistic chaotic sequence. Every second hexadecimal bit is converted into a binary image of the high and low quarters, and the positions of the binary 0 s and 1 s in the high and low quarters are inversely permuted. Then, the high and low quarters in the binary numbers are interchanged to obtain the binary information of the plaintext gray value. Finally, all of the binary is converted to decimals and the three channel images are combined to form the decrypted plaintext image.

#### 3.2.1. Pixel Grayscale Value Inverse Diffusion

This sub-section is the inverse of Section 3.1.5. Pixel gray value inverse substitution is used to recover the magnitude of the changed pixel gray value for the purpose of decryption. In this paper, three types of operations are used: XOR, additive, and subtractive operations. A logistic chaotic sequence *X*1 of length 8MN is intercepted by the first 2MN elements as a new sequence *x*3, which is then quantized to obtain the sequence *X*3, as shown in Equation (11). The Chen chaos sequence is also quantized to obtain the new sequences *X*, *Y*, and *Z*, as shown in Equation (12). the elements in *X*, *Y*, and *Z* are all permutations of integers between 0 and 15. Among them, *X* is used for the inverse diffusion of the pixel gray value for an R-channel image, *Y* is used for the inverse diffusion of the pixel gray value for a g-channel image, and *Z* is used for the inverse diffusion of the pixel gray value for a B-channel image. Take the R channel as an example: if *X*3(*i*) = 0, then perform the XOR operation on Pr6(*i*) and *X*(*i*); if *X*3(*i*) = 1, then perform a subtraction operation on Pr6(*i*) and *X*(*i*); if *X*3(*i*) = 2, then perform an addition operation on Pr6(*i*) and *X*(*i*). The final result is the encrypted image Pr5 after the grayscale inverse diffusion. The calculation is shown in Equation (14):(14)$$\left\{\begin{array}{c} \mathrm{Pr}5\left(i\right)=f_{⊕}\left(\mathrm{Pr}6\left(i\right),X\left(i\right)\right)\;\;\;\;X3=0\\ \mathrm{Pr}5\left(i\right)=f_{-}\left(\mathrm{Pr}6\left(i\right),X\left(i\right)\right)\;\;\;\;X3=1\\ \mathrm{Pr}5\left(i\right)=f_{+}\left(\mathrm{Pr}6\left(i\right),X\left(i\right)\right)\;\;\;\;X3=2 \end{array}\right.\;\;i=1,\;2,\ldots ,\;2\mathrm{MN}$$

#### 3.2.2. Pixel Position Inverse Scrambling

This subsection and Section 3.1.4 are inverse processes to each other. The logistic chaotic sequence *X*2 is subjected to an ascending operation to obtain the index sequence index, and then the index sequence index is used to perform an inverse permutation of Pr5 to obtain an image Pr4 of size 1 × 2MN, as shown in Equation (15): (15)$$\left\{\begin{array}{c} \left[B\;index\right]=sort\left(X2\right)\\ \mathrm{Pr}4\left(index\right)=\mathrm{Pr}5\;\;\; \end{array}\right.\;\;\;\;i=1,\;2,\;3,\ldots ,\;2\mathrm{MN}$$

## 4. Case Study

### 4.1. Experimental Results

The plaintext images selected for this paper are color images of Lena, Mandril, Peppers, and House, which have a size of 512 × 512 × 3. The simulation software MATLAB 2018b is used to encrypt and decrypt the Lena image, and the result is shown in Figure 6. It can be seen from Figure 6 that the encrypted image presents noise-like characteristics and the connection between the encrypted image and the original image is not subjectively visible, indicating that the algorithm has good confidentiality.

### 4.2. Safety Analysis

#### 4.2.1. Key Space Analysis

The size of the key space in an encryption algorithm is an important part of it. The larger the key space, the better the security performance of encryption. In order to prevent brute-force attacks, the size of the key space is at least $2^{100}\approx 10^{30}$. Assuming that the calculation accuracy of the computer is $10^{-15}$, the parameters and initial values of the chaotic system in this paper are μ1, μ2, *a*, *b*, *c*, *X*1(0), *X*2(0), *x*(0), *y*(0), and *z*(0). The key space is at least $10^{150}$, which is much larger than $2^{100}$. Thus, the algorithm can resist brute-force cracking.

#### 4.2.2. Information Entropy Analysis

Information entropy is an important criterion to measure the randomness of information. In a grayscale image, the theoretical value of information entropy is 8. The closer the information entropy is to 8, the stronger the randomness and the stronger the ability to resist attacks. The calculation of information entropy is shown in Equation (16): (16)$$H\left(m\right)=\sum_{i=0}^{2^{l}-1}p\left(m_{i}\right)log_{2}\left(\frac{1}{p\left(m_{i}\right)}\right)$$
where *l* is the number of digits of the image pixel gray value. $\;p\left(m_{i}\right)$ is the probability of the gray level of the pixel. The information entropy of the three images of the R, G, and B channels in the plaintext image and the encrypted image is calculated using Equation (14), as shown in Table 2.

It can be seen from Table 2 that the information entropy of the plaintext image is low, and the information entropy of the encrypted image has been significantly improved. The difference between the information entropy of the encrypted image and the theoretical value is less than 0.001. Compared with other algorithms, the encryption algorithm proposed in this paper has better information entropy, which can show that this algorithm has good random characteristics.

#### 4.2.3. Correlation Analysis

Correlation reflects the connection between adjacent pixels. Due to the high redundancy of image information, the correlation of general images is very high. A good encryption algorithm should reduce the correlation of adjacent pixels, and the ideal value is 0. The calculation correlation is shown in Equation (17):(17)$$\left\{\begin{array}{c} E\left(x\right)=\frac{1}{N}\sum_{i=1}^{N}x_{i}\;\;\;\;\;\;\;\;\;\;\;\;\;\;\;\;\;\;\;\;\;\\ D\left(x\right)=\frac{1}{N}\sum_{i=1}^{N}{\left(x_{i}-E\left(x\right)\right)}^{2}\;\;\;\;\;\;\;\;\;\;\;\;\\ cov\left(x,y\right)=\frac{1}{N}\sum_{i=1}^{N}\left(x_{i}-E\left(x\right)\right)\left(y_{i}-E\left(y\right)\right)\\ r_{xy}=\frac{cov\left(x,y\right)}{\sqrt{D\left(x\right)}\sqrt{D\left(y\right)}}\;\;\;\;\;\;\;\;\;\;\;\;\;\;\;\;\;\;\;\;\; \end{array}\right.$$
where *N* is the number of image pixel pairs. *x*, *y* are the grayscale values of two adjacent pixels, *E*(*x*) is the mean, *D*(*x*) is the variance, and *cov*(*x*,*y*) is the covariance. Equation (17) was used to calculate the image correlation, and the results are shown in Table 3. Experimental simulations of the correlation for the R, G, and B channels are shown in Figure 7, Figure 8 and Figure 9.

It can be seen from Table 3 that the correlation coefficient of the plaintext image is very close to 1, indicating that it has a high degree of correlation and redundancy. The correlation coefficient of the ciphertext image is very close to 0, which can indicate that the algorithm in this paper destroys the correlation between adjacent pixels. Compared with other literature, the algorithm in this paper has lower correlation and can resist statistical attacks well.

It can be seen from Figure 7, Figure 8 and Figure 9 that the gray values of the plaintext image pixels are mainly concentrated on the diagonal, with high density and strong correlation. The pixel gray values of the ciphertext image are evenly dispersed, and the correlation is weak. This can be explained by noting that the encrypted image has low correlation.

#### 4.2.4. Histogram Analysis

The histogram reflects the distribution of the image pixel gray values, and uneven distributions of image pixel gray values are subject to statistical attack. A simulation of the gray histogram for the Lena image’s R, G, and B channel images is shown in Figure 10.

It can be seen from Figure 10 that before the image is encrypted, the grayscale of the three-channel image varies widely, which makes it susceptible to statistical attacks. After the image is encrypted, the gray changes of the three-channel images are relatively uniform, which is beneficial for resisting statistical attacks.

#### 4.2.5. Analysis of Shear Attack

In the process of image information transmission, some information of the image will be lost. The fact that the plaintext information can still be decrypted after some of the information has been lost indicates that the encryption algorithm is well resistant to shear attacks. The experimental simulation results are shown in Figure 11.

As can be seen from Figure 11, when the cipher text is cropped by one-sixteenth, the decrypted image is good. When the cipher text is cropped by one-fourth, the decrypted image is good. When the cipher text is cropped by one-half, the decrypted image can basically restore the main information. Therefore, the algorithm proposed in this paper has good resistance to clipping attacks.

#### 4.2.6. Noise Attack Analysis

To test the performance of encrypted images to resist noise attacks, pretzel noise was added to the encrypted images, and the experimental simulation results obtained after decryption are shown in Figure 12.

Figure 12 shows that, after being affected by noise of different intensities, the decrypted image can still recover the main information of the plaintext, indicating that it has a certain ability to resist noise attacks.

#### 4.2.7. Differential Attack Analysis

A differential attack is a means of cracking an encryption system by changing the pixel gray value of the plaintext and by comparing the relationship between the ciphertext before and after the change. Good encryption algorithms need to be very sensitive to the plaintext information, making large changes to the encrypted image when the plaintext information changes slightly. This is generally measured using the number of pixel changes rate (*NPCR*) and the unified average change intensity (*UACI*), which are calculated using the relevant equations shown in Equation (18): (18)$$\left\{\begin{array}{c} NPCR=\frac{1}{MN}\sum_{i=1}^{M}\sum_{j=1}^{N}D\left(i,j\right)\times 100\%\;\;\;\;\;\;\;\;\;\;\;\\ UACI=\frac{1}{MN}\frac{\sum_{i=1}^{M}\sum_{j=1}^{N}\left|C1\left(i,j\right)-C2\left(i,j\right)\right|}{255}\times 100\%\;\;\;\;\;\;\;\;\\ D\left(i,j\right)=\left\{\begin{array}{c} 1\;\;\;C1\left(i,j\right)\neq C2\left(i,j\right)\\ 0\;\;C1\left(i,j\right)=C2\left(i,j\right) \end{array}\right.\times 100\%\;\;\;\;\;\;\;\; \end{array}\right.$$
where, when the gray value of a pixel value in the plaintext changes, the ciphertext before and after the change is obtained, and the two ciphertexts are *C*1(*i*,*j*) and *C*2(*i*,*j*). M and N are the number of pixels in the rows and columns of the image. The *NPCR* and *UACI*. calculated using Equation (16), are shown in Table 4:

When the NPCR value is greater than 0.996 and the UACI value is between (0.33329, 0.335541), it can pass the test in [27]. The ideal values of *NPCR* and *UACI* are 99.6094% and 33.4635% [28]. It can be seen from Table 4 that the *NPCR* and *UACI* of this algorithm can pass the test of [27]. The test results of *NPCR* and *UACI* in other literature are within the acceptable range. Compared with this algorithm, the values of NPCR and UACI are closer to the theoretical value. Therefore, the encrypted image obtained by this algorithm is very sensitive to the plaintext information, which is beneficial for resisting differential attacks.

#### 4.2.8. Computational Complexity

Computational complexity is an important metric for measuring encryption algorithms. However, encryption and decryption change depending on the system configuration and the hardware and software of the computer, so it is not reasonable to compare with existing algorithms. Thus, it can be measured by the time complexity of the encryption process. The Chen chaos system is a three-dimensional chaos system that generates chaotic sequences with time complexity Θ (3 × MN). The first logistic chaos sequence is used to internally permute the bit plane with time complexity Θ (8 × MN). The second logistic chaos sequence is used for position scrambling with time complexity bits Θ (2 × MN), and the Chen chaos sequence is used for the permutation of pixel gray values with a time complexity of bit Θ (2 × MN). Therefore, the time complexity is Θ (8 × MN). The time complexity comparisons are shown in Table 5. By comparison, we will find that the time complexity in [20] is the same as this paper, but the key space in [20] is $10^{64}$, and the key space in this paper is $10^{150}$. Thus, this paper has a larger key space and better security performance. This paper is compared with [32] and the time complexity is lower than it. Therefore, in the case of good security, the algorithm has low complexity.

## 5. Discussion

This paper proposes a low-complexity color image encryption algorithm based on a combination of bit-plane and chaotic systems interlinked with plaintext information. Improvements are made to the deficiencies in the algorithms of references [17,18,19,31]. The process of encryption in [17] does not encrypt the lower-four bit planes and may be subject to selected plaintext attacks. In order to overcome this deficiency, the pixel bit-plane internal scrambling algorithm is introduced in this paper, which encrypts not only the higher-four bit planes but also the lower-four bit planes, while the positions of the higher- and lower-four bits are swapped, making the algorithm more resistant to selective plaintext attacks. In [18], the selection of the initial value of the chaotic system relies on the input of an external key, which cannot vary with the image and may be subject to the chosen plaintext attack. To overcome this shortcoming, the algorithm in this paper introduces an algorithm where the key is related to the plaintext information. The initial value of a chaotic sequence is correlated not only with its corresponding channel, but also with information from other channel images. When a pixel value of a channel image changes, not only the chaotic sequence encrypting the current channel, but also the chaotic sequence encrypting the images of the other two channels changes, making it more resistant to a selective plaintext attack. The authors of [19] can only disrupt the positions between two bit planes at the same time for each permutation operation, and cannot disrupt the positions of bits between multiple bit planes at the same time, which may be subject to selective plaintext attacks. To overcome this shortcoming, the pixel bit-plane internal scrambling algorithm is introduced in this paper’s algorithm, which can displace four bit-planes at the same time, making the resistance to selective plaintext attacks stronger. The computational complexity of the encryption algorithm in [31] is high. In order to overcome this shortcoming, this paper uses the Chen chaos system for encryption, which reduces the complexity and has a better encryption effect. The computational complexity is Θ (24 × MN) in [31] and the computational complexity is Θ (8 × MN) in this paper. Finally, some improvements are made to address some of the shortcomings of the above literature, so that the algorithm proposed in this paper has high security properties and relatively low complexity.

## 6. Conclusions

This paper encrypts color images based on the bit-plane information of the image and the Chen chaos system. By partitioning the bit plane, the computational complexity of the encryption process can be reduced. The choice of key space is related to the plaintext information of the image. Different images have different keys, and when the information of different channels changes, it will also affect the key of other channels’ image encryption, which increases the security of the image. To prove the security, this paper analyses the security of the encryption algorithm in terms of key space, information entropy, correlation, histogram, clipping attacks, noise attacks, and difference attacks. Simulation experiments show that the algorithm has a good encryption effect and is resistant to attacks. The chaotic system used in this paper is a classical chaotic system, so the use of this chaotic system for image encryption is easy to reproduce and can be easily studied by later researchers. The encryption method can be applied to medical images that require high security and low complexity.

## Figures and Tables

**Figure 1 entropy-24-00186-f001:**
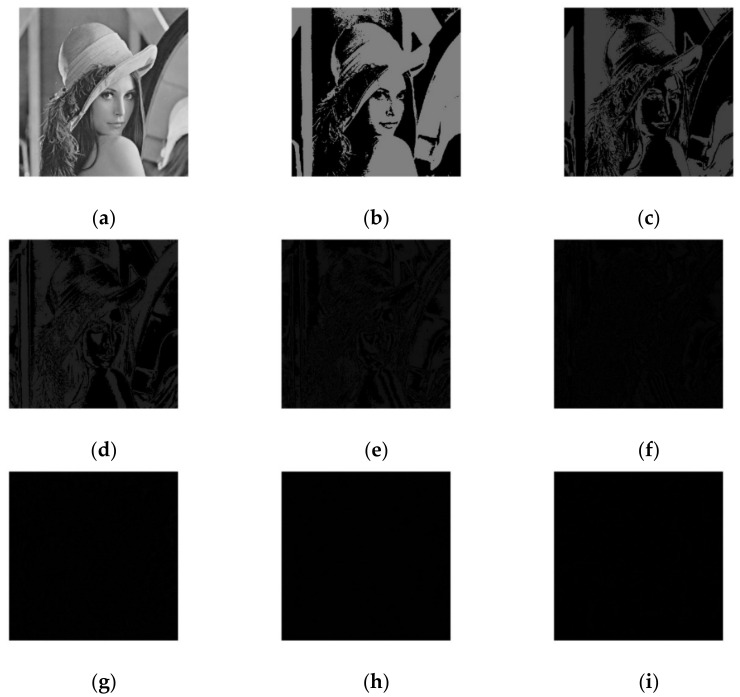
Bit-plane infographic of Lena image. (**a**) Lena image; (**b**) The 8th plane of Lena image; (**c**) The 7th plane of Lena image; (**d**) The 6th plane of Lena image; (**e**) The 5th plane of Lena image; (**f**) The 4th plane of Lena image; (**g**) The 3rd plane of Lena image; (**h**) The 2nd plane of Lena image; (**i**) The 1st plane of Lena image.

**Figure 2 entropy-24-00186-f002:**
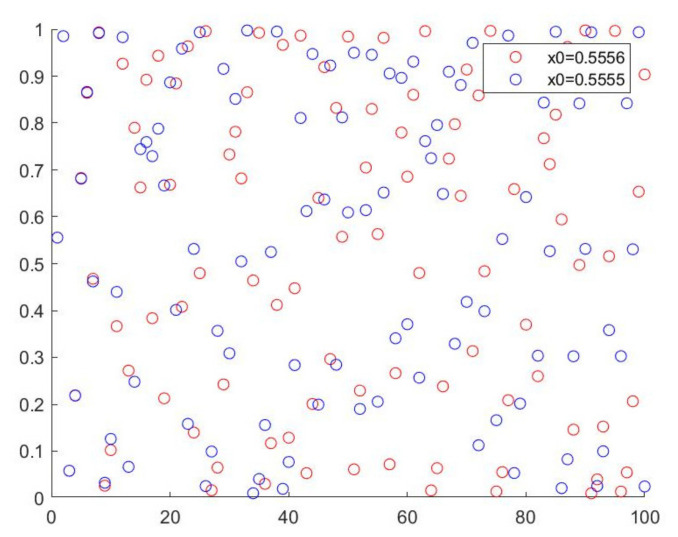
Scatter plot of a logistic chaotic sequence.

**Figure 3 entropy-24-00186-f003:**
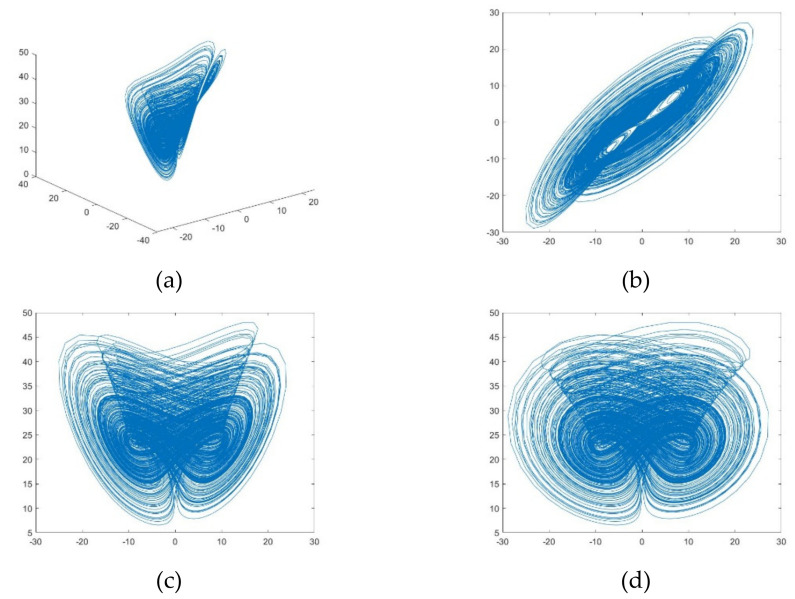
Attractor of the Chen chaotic system. (**a**) Three-dimensional view of the Chen chaotic system; (**b**) The *x*-*y* plane diagram of the Chen chaotic system; (**c**) The *x*-*z* plane diagram of the Chen chaotic system; (**d**) The *y*-*z* plane diagram of the Chen chaotic system.

**Figure 4 entropy-24-00186-f004:**
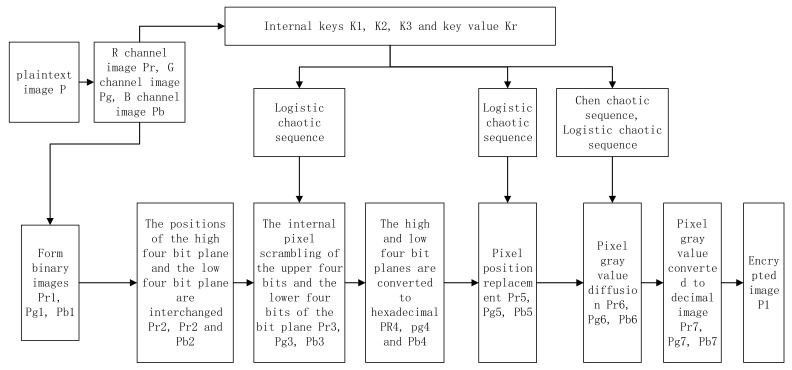
Block diagram of the encryption algorithm.

**Figure 5 entropy-24-00186-f005:**
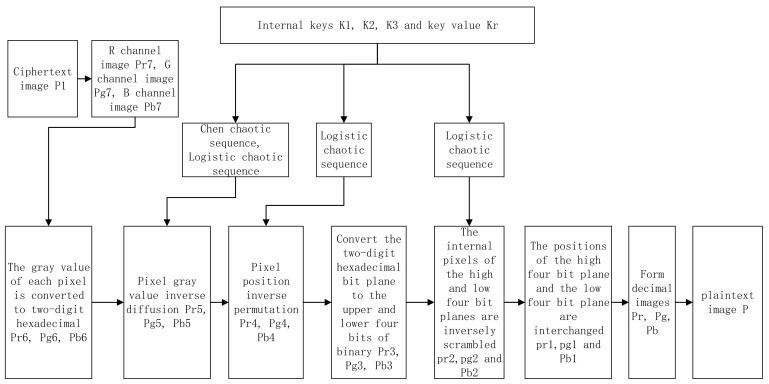
Block diagram of the decryption algorithm.

**Figure 6 entropy-24-00186-f006:**
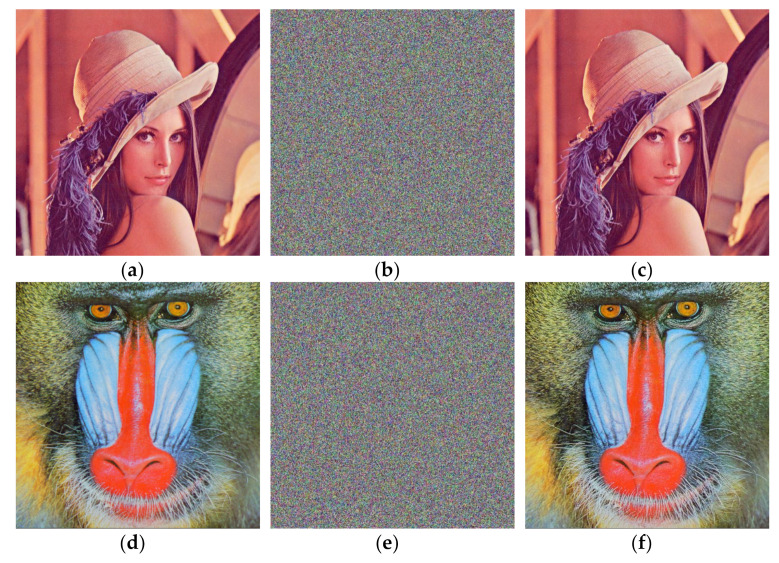
Image encryption and decryption results. (**a**) Lena original image; (**b**) Lena encryption image; (**c**) Lena decryption image; (**d**) Mandril original image; (**e**) Mandril encryption image; (**f**) Mandril decryption image; (**g**) Peppers original image; (**h**) Peppers encryption image; (**i**) Peppers decryption image; (**j**) House original image; (**k**) House encryption image; (**l**) House decryption image.

**Figure 7 entropy-24-00186-f007:**
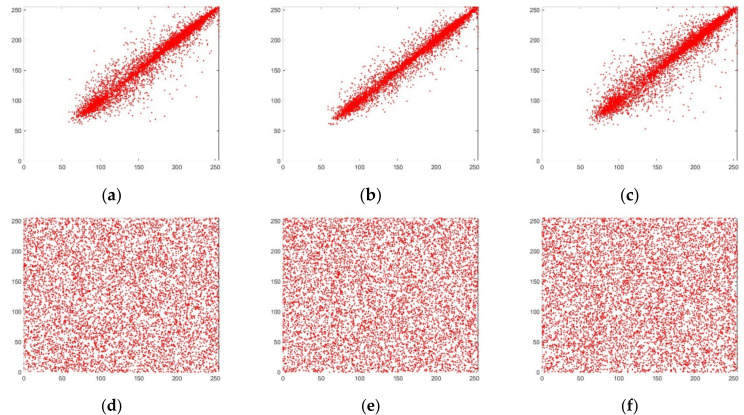
R-channel correlation analysis. (**a**) R-channel plaintext horizontal correlation; (**b**) R-channel plaintext vertical correlation; (**c**) R-channel plaintext diagonal correlation; (**d**) R-channel ciphertext horizontal correlation; (**e**) R-channel ciphertext vertical correlation; (**f**) R-channel ciphertext diagonal correlation.

**Figure 8 entropy-24-00186-f008:**
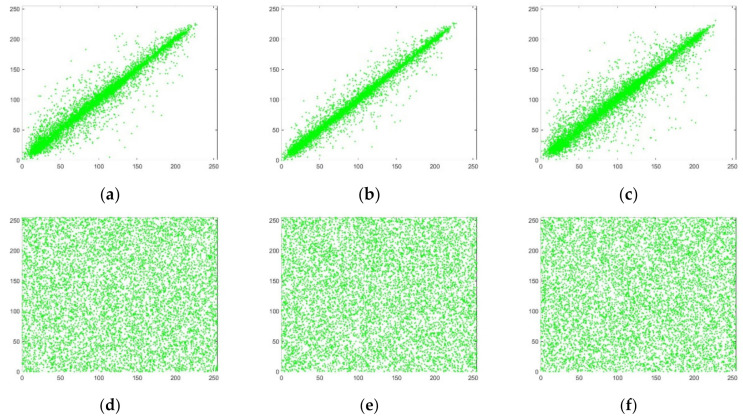
G-channel correlation analysis. (**a**) G-channel plaintext horizontal correlation; (**b**) G-channel plaintext vertical correlation; (**c**) G-channel plaintext diagonal correlation; (**d**) G-channel ciphertext horizontal correlation; (**e**) G-channel ciphertext vertical correlation; (**f**) G-channel ciphertext diagonal correlation.

**Figure 9 entropy-24-00186-f009:**
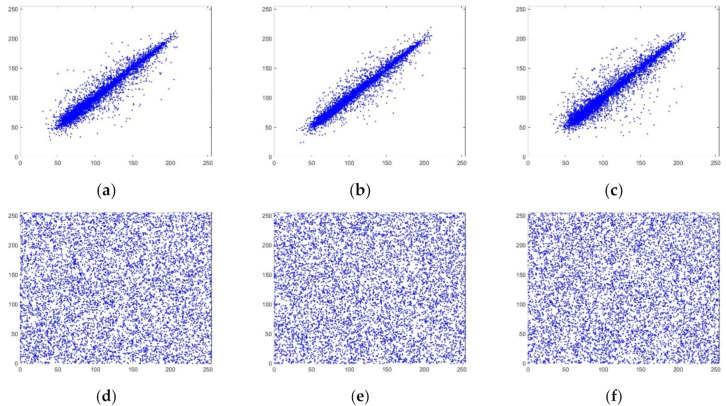
B-channel correlation analysis. (**a**) B-channel plaintext horizontal correlation; (**b**) B-channel plaintext vertical correlation; (**c**) B-channel plaintext diagonal correlation; (**d**) B-channel ciphertext horizontal correlation; (**e**) B-channel ciphertext vertical correlation; (**f**) B-channel ciphertext diagonal correlation.

**Figure 10 entropy-24-00186-f010:**
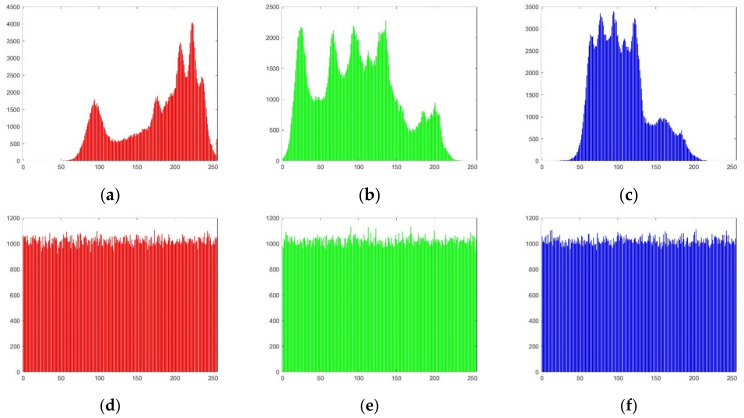
Histogram analysis. (**a**) Plaintext R-channel histogram; (**b**) Plaintext G-channel histogram; (**c**) Plaintext B-channel histogram; (**d**) Ciphertext R-channel histogram; (**e**) Ciphertext G-channel histogram; (**f**) Ciphertext B-channel histogram.

**Figure 11 entropy-24-00186-f011:**
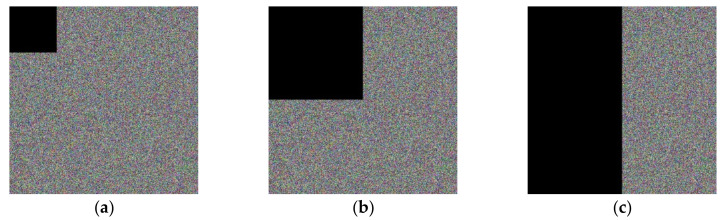
Decryption results of ciphertexts with different cropping ratios. (**a**) Ciphertext cropping 1/16; (**b**) Ciphertext cropping 1/4; (**c**) Ciphertext cropping 1/2; (**d**) Decrypted image after cropping 1/16; (**e**) Decrypted image after cropping 1/4; (**f**) Decrypted image after cropping 1/2.

**Figure 12 entropy-24-00186-f012:**
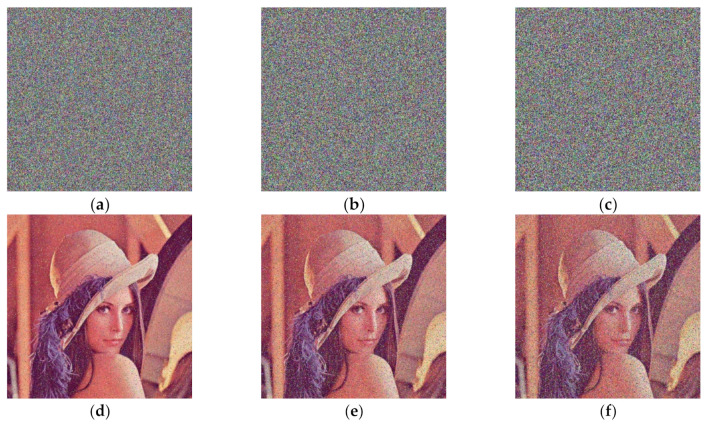
Encrypted image and decrypted image after adding noise. (**a**) Add 0.1 salt-and-pepper noise; (**b**) Add 0.2 salt-and-pepper noise; (**c**) Add 0.3 salt-and-pepper noise; (**d**) 0.1 salt-and-pepper noise decryption image; (**e**) 0.2 salt-and-pepper noise decryption image; (**f**) 0.3 salt-and-pepper noise decryption image.

**Table 1 entropy-24-00186-t001:** Weights and proportions of pixel bit planes.

Bit Plane	8	7	6	5	4	3	2	1
Weight	$2^{7}$	$2^{6}$	$2^{5}$	$2^{4}$	$2^{3}$	$2^{2}$	$2^{1}$	$2^{0}$
Proportion	50.196%	25.098%	12.549%	6.275%	3.137%	1.569%	0.784%	0.392%

**Table 2 entropy-24-00186-t002:** Information entropy.

Image	Plaintext Information Entropy		Ciphertext Information Entropy		
			This Paper’s Algorithm	Reference [23]	Reference [24]
Lena	R channel	7.2682	7.9993	7.9992	7.998
	G channel	7.5901	7.9993	7.9993	7.998
	B channel	6.9951	7.9994	7.9992	7.997
Mandrill	R channel	7.7067	7.9994	7.9993	7.997
	G channel	7.4744	7.9994	7.9993	7.997
	B channel	7.7522	7.9993	7.9993	7.997
Peppers	R channel	7.3388	7.9993	7.9993	7.998
	G channel	7.4962	7.9993	7.9993	7.997
	B channel	7.0583	7.9993	7.9992	7.997

**Table 3 entropy-24-00186-t003:** Color image correlation.

Image		Horizontal	Vertical	Diagonal
Lena plaintext image	R	0.9771	0.9889	0.9668
	G	0.9796	0.9875	0.9681
	B	0.9517	0.9731	0.9269
Lena—our method	R	0.0040	−0.0012	0.0113
	G	−0.0013	0.0079	0.0037
	B	0.0025	−0.0007	0.0021
Mandrill plaintext image	R	0.9139	0.8561	0.8491
	G	0.8778	0.7862	0.7543
	B	0.9153	0.9025	0.8637
Mandrill—our method	R	−0.0028	0.0003	−0.0141
	G	0.0044	−0.0043	−0.0110
	B	−0.0062	−0.0116	−0.0003
Peppers plaintext image	R	0.9637	0.9695	0.9647
	G	0.9869	0.9890	0.9820
	B	0.9625	0.9700	0.9508
Peppers—our method	R	−0.0104	0.0178	−0.0049
	G	0.0180	0.0006	−0.000015
	B	0.0087	−0.0063	0.0003
House plaintext image	R	0.9572	0.9476	0.9149
	G	0.9584	0.9606	0.9262
	B	0.9746	0.9788	0.9568
House—our method	R	0.0013	−0.0009	−0.0145
	G	0.0069	0.0193	0.0004
	B	0.0083	−0.0057	−0.0016
Lena Reference [25]	R	−0.0064	0.0053	0.0061
	G	0.0018	−0.0047	0.0027
	B	0.0099	0.0043	0.0035
Mandrill Reference [26]	R	−0.0058	0.0014	−0.0027
	G	−0.0023	−0.0082	0.0090
	B	−0.0068	0.0065	−0.0091
Peppers Reference [12]	R	0.0087	−0.0063	0.0003
	G	−0.0112	−0.0076	−0.0028
	B	−0.0060	−0.0021	0.0018

**Table 4 entropy-24-00186-t004:** NPCR and UACI calculation of three channels of RGB image.

Methods	Color Image Channel	NPCR (%)	UACI (%)
Lena—our method	R	99.6136	33.4783
	G	99.5922	33.4769
	B	99.6109	33.4916
Mandrill—our method	R	99.5991	33.4989
	G	99.6120	33.4906
	B	99.5785	33.4114
Peppers—our method	R	99.5865	33.5472
	G	99.6052	33.3542
	B	99.6094	33.4330
House—our method	R	99.6029	33.4591
	G	99.6117	33.5023
	B	99.6124	33.5843
Lena Reference [29]	R	99.6016	33.2483
	G	99.6205	33.4977
	B	99.6095	33.3877
Lena Reference [30]	R	99.6056	33.4108
	G	99.6147	33.4653
	B	99.6235	33.4901
Lena Reference [31]	R	99.6096	33.4926
	G	99.6102	33.4620
	B	99.5921	33.4961

**Table 5 entropy-24-00186-t005:** Computational complexity.

Algorithm	Computational Complexity
Our method	Θ (8 × MN)
Reference [20]	Θ (8 × MN)
Reference [32]	Θ (24 × MN)

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
