# Peer review of "Research on Color Image Encryption Algorithm Based on Bit-Plane and Chen Chaotic System"

_entropy, 2022, doi:10.3390/e24020186_

Round 1

Reviewer 1 Report

This manuscript has proposed a novel technique to encrypt RGB images. The proposed image encryption technique has the following six steps:

1. The high and low four bits of the binary gray value of each image pixel are exchanged.

2. The position of each four-bit number is scrambled by Logistic chaotic sequence. 

3. The four-bit numbers are converted to hexadecimal numbers.

4. The position of scrambled is transformed by Logistic chaotic sequence.

5. Chen chaotic sequence replaces the pixel's gray value of the scrambled image.

6. Finally, the encrypted image is into a decimal number to form a single channel encrypted image. 

 Authors have extensively discussed the mathematical steps used in image encryption. The results of the proposed encryption method have been clearly presented in the results section of the paper, where the proposed encryption method has been compared with earlier methods. 

I have some concerns regarding the manuscript that I strongly feel need to be addressed. 

  1. The authors have shown the mathematical formulation of this algorithm of "Chen chaotic sequence" in section 2.2.2, but they have not cited the prior work which proposed this equation. Since the "Chen chaotic sequence" is a critical mathematical formulation in the proposed encryption steps, the authors must add a citation of this prior research.
    If there is no such prior work, the authors should prove why this equation is appropriate for the proposed encryption technique. 
  2. In the manuscript's abstract, the authors have claimed that the proposed encryption technique is low complexity and has small calculations (lines 11-12). How did the authors come to this conclusion? What benefit is achieved in the proposed encryption method compared to prior methods? 
  3. The authors have described the decryption algorithm in a single paragraph in Section 4. So that the researchers can reproduce the results that the authors have shown in the paper, the authors should explain the decryption algorithm in the same detail as the encryption algorithm. 
  4. In Section 5, Tables 2, 3, and 4, the authors have compared their method with the prior work cited in references. It is not clear what benefit the proposed work achieves compared with previous research. As pointed in point 2, the authors should show through experiments that the proposed encryption research achieves low complexity and less number of calculations when compared to the references [6], [12], [13], [14], [15], [16], [17], [18] and [19]. Otherwise, all of the analysis results suggest no benefit of using the proposed research. 
  5. Authors have focused their encryption analysis on just one image (Lena). To show the potential of the proposed research, authors should extend their analysis to more diverse standard image processing images. For example, authors can use the image of 
    1. Mandrill (https://sipi.usc.edu/database/database.php?volume=misc&image=10#top)
    2. Peppers (https://sipi.usc.edu/database/database.php?volume=misc&image=13#top)
    3. House (https://sipi.usc.edu/database/database.php?volume=misc&image=37#top)

Apart from the significant revisions suggested above, authors should also look at a few minor corrections that might be required in the paper.

  1. Typo in line 322 (page 11 of 13). "Where," has been written twice. 
  2. At the far right end of line 323, there is a strange character "å’Œ". Perhaps this got introduced when the pdf file got created. 
  3. It would benefit the research community if the authors can post their MATLAB code on GitHub. This way, other researchers can reproduce the proposed research and extend the field even further. 

Reviewer 2 Report

After reading the paper, as a research or scientific manuscript, I find this paper problematic. Although there are certainly merits in this work, paper needs great revision. In its current form, I remain skeptical of the implications for academics and practitioners of paper findings.

In fact, it looks to me more like a technical report. In a scientific paper the authors have to discuss their work in context of related work and they have to elaborate what the original contribution to the state of the art is. Unfortunately, this paper fails completely in all of these aspects. As a research paper, the paper needs to be improved. The literature needs to be integrated with the claims that the authors make in order to show the importance of their contribution. This is a common problem I found in each section.

Introduction: The justification for the motivation of this study is relatively weak and not clearly explained. Introduction should show paper motivation, paper purpose and which is the paper knowledge contribution. After reading it, the research objectives and their importance are hidden. In addition, there should be clearly defined research questions.

Theoretical background/Literature review. Paper reads like an internal report instead of an academic research paper. In fact, it looks to me more like a technical report of a case study. It does not discuss alternative approaches, and it does not discuss weakness and strengths. In a scientific paper the authors have to discuss their work in context of related work and they have to elaborate what the original contribution to the state of the art is. Unfortunately, this paper fails completely in all of these aspects. The literature review (section 2 basic theory) is short and the results should be presented in a different way.

The research method of the case study and the research method followed to obtain author’s proposal as well is not well justify, so we don’t know if the results are correct or not.

Section 4. Decryption algorithm. A section with only one paragraph has no sense

A Discussion section is necessary. The contribution of the author’s approach to the literature is not highlighted. The literature review needs to be integrated with the claims that the author make in order to show the importance of its contribution. The piece is lacking in originality or a clear contribution to the literature.

 Conclusion section: I would like that authors show better the consequences for academics and practitioners of the results. It has to be showed in the conclusion section

In conclusion, I believe the proposal has some merit, but I do not believe the paper in its current form demonstrates this potential as a research paper

Round 2

Reviewer 1 Report

Authors have addressed all of my prior comments. I am comfortable in marking the current version of manuscript as accept in its present form. 

Reviewer 2 Report

Paper has been improved and I find merit in it. I would recommend the following minor changes:

  1. Introduction: A small paragraph acting as a reader's guide should be featured at the end of the introduction
  2. Sections name. I think that paper would benefit if authors use classical section names. Therefore, I would structure this paper in introduction, literature review, research method (the research method used to develop the algorithm not the proposed algorithm), proposed algorithm, case study, discussion, conclusion. It would help readers to understand the paper
  3. Conclusion section: I would like that authors show better the consequences for academics and practitioners of the results. It has to be showed in the conclusion section
